# The Role of Peptides in Nutrition: Insights into Metabolic, Musculoskeletal, and Behavioral Health: A Systematic Review

**DOI:** 10.3390/ijms26136043

**Published:** 2025-06-24

**Authors:** Syed Khuram Zakir, Bilal Jawed, Jessica Elisabetta Esposito, Rimsha Kanwal, Riccardo Pulcini, Riccardo Martinotti, Edmondo Ceci, Matteo Botteghi, Francesco Gaudio, Elena Toniato, Stefano Martinotti

**Affiliations:** 1Center of Advanced Studies and Technology, Department of Innovative Technology in Medicine and Dentistry, G. d’Annunzio University, 66100 Chieti, Italy; khuramabbas512@gmail.com (S.K.Z.); bilaljawed2007@gmail.com (B.J.); j.elisabetta.esposito@gmail.com (J.E.E.); kanwalrimsha809@gmail.com (R.K.); riccardo.pulcini@unich.it (R.P.); 2Unit of Clinical Pathology and Microbiology, Miulli Generale Hospital, LUM University, 70021 Acquaviva delle Fonti, Italy; martinotti@lum.it; 3Residency Program in Clinical Pathology and Clinical Biochemistry, Department of Applied Clinical and Biotechnological Science, University of L’Aquila, 67100 L’Aquila, Italy; 4Residency Program in Clinical Oncology, Faculty of Medicine, Umberto I University Hospital, University of Rome “La Sapienza”, 00185 Rome, Italy; riccardo.martinotti@uniroma1.it; 5Veterinary Medicine Department, University of Bari “Aldo Moro”, 70010 Bari, Italy; e.ceci@miulli.it; 6Experimental Pathology Research Group, Department of Clinical and Molecular Sciences, Università Politecnica delle Marche, 60126 Ancona, Italy; matteo.botteghi@worldconnex.com; 7Unit of Haematology, Department of Medicine and Surgeon, F. Miulli University Hospital, Libera Università Mediterranea (LUM) “Jean Monnet” Casamassima, 70010 Bari, Italy

**Keywords:** peptides, bioactive peptides, metabolic health, musculoskeletal function, GLP-1 nutritional supplements

## Abstract

Peptides are currently vital components in nutrition with physiological advantages beyond a basic diet. This systematic review aims to explain their significance in metabolic, behavioral, and musculoskeletal health, focusing on their therapeutic benefits, molecular mechanisms, and bioactivities. This systematic review analyzed clinical trials from PubMed and Scopus databases in the time range of 2019 to 2024, following the Preferred Reporting Items for Systematic Reviews and Meta-Analyses (PRISMA) standards, that investigated the role of peptides in human nutrition. Eight randomized clinical trials (RCTs) met the predefined metabolic, behavioral, and musculoskeletal health inclusion criteria. Peptides are derived from various sources, including milk, fish, and plants, and show various bioactive characteristics such as anti-inflammatory effect, improved muscle protein synthesis, and immune modulation. Some important findings emphasize their potential to govern metabolic processes, defend against chronic diseases, and enhance gut health. For instance, glucagon-like peptide (GLP-1) controls taste perception and appetite stimulation, and collagen peptides strengthen the musculoskeletal system. Peptides display intriguing potential as nutrients for addressing global health challenges, including behavioral responses, aging, and metabolic syndrome. Future investigations would focus on bioavailability, optimizing dosage, and demographic-specific treatments.

## 1. Introduction

In the last few decades, peptides have evolved as critical constituents in nutrition since they provide physiological gains above and beyond meeting basic dietary requirements [1,2]. Peptides are short chains of amino acids, typically consisting of 2 to 50 residues joined by covalent bonds formed during condensation reactions [3]. Most peptides commonly used as supplements are found in several nutrients, such as marine organisms (e.g., fish collagen peptides), dairy products (e.g., milk casein hydrolysates), and plant proteins (e.g., rice bran peptides). The activity of these peptides varies by several constituents and the sequencing of amino acids [4,5,6,7,8,9]. The short chains of amino acids are essential for many biochemical and physiological processes [10,11]. Since the last decade, these peptides have received special attention for their physiological activities and have been acknowledged as beneficial for human health. Peptides as bioactive compounds have been illustrated to affect immune system modulation and muscle protein synthesis [12,13].

Recent studies have indicated that some food-derived peptides are beneficial for the management and prevention of chronic diseases like diabetes, osteoarthritis, and hypertension [14,15,16,17,18,19,20,21]. Milk-borne peptides have a variety of bioactive peptides that cause physiological effects such as opioid, antioxidative, antimicrobial, and cytomodulatory [22,23,24]. For instance, peptides derived from fish hydrolysate protein can boost neuropeptide release and stimulate metabolic regulation [25,26].

Moreover, peptides have notable contributions to human health, and their constituents have some advantages in aging and sports nutrition. For example, collagen peptide strengthens older men’s muscles when it binds with resistance training (RT) [27,28,29]. Recent developments in biotechnology have been made by synthesizing peptides that trigger a specific function. As a result, new peptide-based supplements are generated, which give the most efficient outcomes. Peptides originating from casein induce antihypertensive attributes, while collagen peptides are widely used to enhance skin and joint health [30]. Furthermore, glucagon-like peptide (GLP-1) is also synthesized to regulate appetite, which is essential in glucose homeostasis [31].

It has been reported that collagen peptides can enhance daily living (ADLs) activities, improve physical and mental health, and reduce pain in middle-aged adults. This analysis marginally emphasizes prospects for more significant advantages among specific groups, such as high-frequency workouts and females [32,33,34]. Moreover, marine-originated peptides are getting more attention for their role in the synthesis of novel therapeutic agents because of their distinct morphology [35,36,37,38,39,40]. By altering their significant biochemical routes, these bioactive peptides have revealed assurance in the struggle against diabetes, obesity, and other metabolic diseases [41,42,43,44,45]. Some exciting research has reported that taste perception hits differently in males and females [46]. Glucagon-like peptide (GLP-1) has a greater tendency to change taste perception in women than in men. This analysis shows some significant understanding of physiological changes in response to the eating habits of both men and women [47,48,49,50]. The considerable advantages of peptides in nutrition are followed by some challenges with their bioavailability and stability [51,52]. So, it is crucial to understand the action mechanism, proper dosage, and long-lasting consequences. This systematic review aims to assemble the most recent findings addressing the backgrounds, physiological implications, therapeutic uses, and molecular mechanisms of peptides in nutrition.

## 2. Methods

### 2.1. Searching for Data

In April 2025, following a review of the literature, we conducted a systematic evaluation under the Preferred Reporting Items for Systematic Reviews and Meta-Analyses (PRISMA) [53]. The PubMed and Scopus databases were used for the search. Peptides, nano-peptides, short peptides, biopeptides, bioactive peptides, food-derived peptides, nutrition, diet, and supplements-related keywords and phrases found in the title or abstract were used to establish the search methodology for the database. To find studies in the database, we used the connective operators “AND” and “OR” for words like “Peptides” and “Nutrition”. A well-defined list of these keywords and search operators used for the database is presented in Table 1. The objective of our review was to understand the subject by focusing on findings explicitly about the uses of peptides in nutrition to promote health and prevent diseases (Appendix A).

### 2.2. Selection Criteria

Firstly, articles that involved the role of peptides in human nutrition were included. As exhibited in Table 2, using a duration filter, we ensured that the most recent articles from 2019 onwards were included to capture the most recent developments in this field. Only clinical and randomized controlled trials (RCTs) were considered. This study involved human subjects of all ages, encouraging comprehensive analysis across all life stages. To uphold coherence and legibility, we include articles only published in English. All research involving animals, abstracts, case reports, sub-populations with some pathologies, and studies that did not identify the nutritional impact of peptides were also excluded.

### 2.3. Research Screening

After numerous searches, many studies were found and put through a detailed screening process to ensure their impact and quality. Two independent reviewers evaluated the title and abstract, and then the full texts were screened against preset inclusion and exclusion criteria. Data from defined studies were retrieved using a standard sheet to ensure quality and suitable insights into the significance of peptides in nutrition. The PRISMA flow diagram (Figure 1) shows the breakdown of the literature search results.

### 2.4. Risk of Bias Assessment

A risk of bias assessment was conducted using the Cochrane RoB 2.0 tool for all selected studies [54]. All trials were rated as low risk in significant domains like randomization process (D1), deviation from intended interventions (D2), missing outcome data (D3), and measurements of outcomes (D4). However, all these investigations indicated “some concern” in selecting the reported studies’ domain (D5) due to the absence of adequate transparency and predefined procedures Figure 2.

## 3. Results

A total of 17,325 articles from the PubMed and Scopus databases were found during the initial search. After filtering these articles by their abstracts and titles, 16,575 articles were left out, and 750 articles were identified after we narrowed down our investigation to include only articles published since 2019. These 750 articles went through full-text screening, resulting in the inclusion of 458 articles. Out of these 458 articles, only nine articles were close to meeting the full criteria for the systematic review. Subsequently, one more article was left out as it does not show relevance to our ultimate research aim. In the end, we found eight articles that satisfied our specifications and were taken into consideration for our review (Figure 1).

### 3.1. Study Layout

The first study comprised double-masked, randomized, and crossover trials involving 20 healthy adults (Aged 26 ± 7). Each experiment inspected energy expenditure, blood pressure, and appetite to examine the effect of various calcium sources and protein co-ingestion on postprandial glucagon-like-peptide (GLP-1), glucose-dependent insulinotropic polypeptide (GIP), and peptide tyrosine–tyrosine (PYY) responses as shown in Table 3 [55]. In the second study, 120 overweight men (Aged 30–60) were allocated randomly to three groups, each receiving a placebo, whey protein, and collagen peptide. As summarized in Table 3, a double-blind, monocentric, and placebo-controlled RCT studied the effects of a 15 g collagen peptide associated with 12 weeks of a resistance training program on muscle strength and body composition in untrained middle-aged men [56]. In the third investigation, 86 participants (aged 40–65) were involved. In a placebo-controlled RCT, double-blind adults were analyzed for pain, function, and physical and mental health throughout 3, 6, and 9 months of supplementation with 0 g, 10 g, and 20 g of collagen peptide per day as demonstrated in Table 3 [57].

In the fourth study (Table 3), which was double-masked, placebo-controlled, and randomized, 25 participants (aged 40–65 years) were treated with 15 g collagen peptide (CP) daily and 12 weeks of hypertrophic resistance exercise to analyze the effects on the skeletal muscle proteome [58]. Two more studies on musculoskeletal health assessment have been reported in Table 3. Genc et al. [59] investigated the effect of collagen in 32 individuals (aged 35–65 years) over 8 weeks on Kinesiophobia, pain, and physical function of individuals with meniscopathy. Meanwhile, Jerger et al. [60] examined 50 active males (aged 18–40) to assess adaptation in the patellar tendon by exercising a 14-week collagen diet and physical exercises [60]. In another crossover, randomized, and placebo-controlled analysis,14 healthy participants (six males, eight females), aged 24.2 ± 2.6, were examined. The effect of GLP-1 on taste preference was examined. The impact of GLP-1 on taste preferences was measured using food illustrations followed by oral sodium load, and all participants received either a GLP-1 or placebo infusion [61]. In the final study listed in Table 3, Jensen et al. [62] carried out a randomized placebo-controlled investigation involving 195 adults (aged 18–65 years) to evaluate the effect of exercise and liraglutide on weight maintenance. This study evaluates weight regain after stopping one year of liraglutide supplements.

### 3.2. Dosage

In the first analysis, each of the 20 participants received one of three supplements: 3745 mg of milk mineral, which contained 2050 mg of calcium and 58.8 g of protein; 4380 mg of calcium citrate, providing 1000 mg of calcium; and 453 mg of protein-rich milk mineral [55]. In the second study, all participants were assigned to receive 15 g (dissolved in 250 mL water) of whey protein, silicon dioxide-containing placebo, or collagen peptides (CP) daily [56]. In the third study, participants were divided into three groups. First, two groups received 10 and 20 g/day of collagen peptides, while the third group got maltodextrin equivalent to the collagen peptide dose. Each individual in the fourth study got 15 g of collagen peptides or a noncaloric placebo [57,58]. All dosages were taken one hour before the training session to ensure a consistent daily intake. In one study, the collagen group received one sachet of Naturengen 4 Joint containing 2000 mg of Type I/III collagen peptides, while the other placebo group was treated with the same sachet containing maltodextrin for 8 weeks [59]. In another study listed in Table 3, the intervention group was treated with 5 g/day of specific collagen peptide. In comparison, participants in the placebo group received 5 g/day of maltodextrin (placebo) with resistance training [60]. In another study, participants received 500 mL of placebo and 1.5 pmol/kg/min GLP-1 every three hours [61]. In the final analysis, the liraglutide group was treated with 0.6 mg/day of liraglutide injection for 52 weeks, while the other group received the same dose of inert placebo, not any other active drug [62].

### 3.3. Measuring Parameters

In this systematic review, peptide sources, physiological effects, and bioavailability are vital parameters for measuring peptide supplementation. In the first study, an inspection of GLP-1 concentration in plasma post-ingestion is measured as the incremental area under the curve (iAUC) [55]. To study physiological effects like hormonal levels of GLP-1, GPY, and PYY using ELISA, appetite is measured via a visual analog scale (VAS). The second analysis anthropometrically investigated physiological parameters such as creatine kinase, urea levels, and body weight. Meanwhile, body composition was measured by Dual Energy X-ray Absorptiometry (DXA) [56]. In the third study, the physical activity survey, the veteran Rand 12-item Healthy Survey (VR-12), food record, Knee injury, and Osteoarthritis Outcomes Score are considered for measuring parameters [57].

In the fourth study, body mass, fat-free mass, and fat mass were measured by the Bioelectrical Impedance Analysis System. A dynamometer was used to calculate leg extension maximal voluntary isometric contraction (MViC), and Liquid Chromatography-Mass Spectrometry (LC-MS) was introduced for measuring muscle proteome analysis [58]. In the fifth study, physical function tests, such as Timed Up and Go (TUG), a 6 min walk, the Berg balance scale (BBS), and a stair-climbing Test, were carried out. The Tampa Scale is used for Kinesiophobia (TSK). The Visual Analog Scale (VAS), WOMAC total, KOOS-PS, and Foot Function Index (FFI) were used to measure pain, quality of life, and physical function [59]. In another investigation, Magnetic resonance imaging (MRI) was used to measure the patellar tendon and the rectus femoris muscle Cross-sectional Area (CSA). The stiffness testing evaluated the patellar tendon’s stiffness. 1-repetition maximum (1 RM) tests assess maximal muscle strength [60].

In another study, body composition analysis was performed by a Bioelectrical Impedance Analyzer. A digital calculator, the Homeostasis Model Assessment (HOMA), was applied to estimate insulin resistance and sensitivity. Finally, the ELISA ALPCO kit was used to assess the GLP-1 level [61]. In the final study, Dual-Energy X-ray Absorptiometry (DXA) is used to measure body weight, and a cycle ergometer calculates VO2 max, and accelerometers were used to assess physical activity level [62]. Table 3 systematically provides the aforementioned measuring parameters explicitly linked to their respective source study.

## 4. Discussion

Peptides have gotten significant attention in nutrition because of their numerous advantages in multifunctional health and capacity to address global health issues. Such peptides are derived from various dietary sources such as plant proteins, meat, fish, and dairy [1,2,3,4,5,6,7,8,9]. These peptides are liberated through gastrointestinal digestion, fermentation, or enzymatic hydrolysis and have diverse biological operations, including antibacterial, antioxidative, immunomodulatory, and antihypertensive effects [2,10,11,12]. Their capacity to regulate metabolic pathways and interact with gut microbiota shows their relevance in enhancing general health, preventing chronic disease, and even contributing to customized diets [63,64]. Despite these optimistic attributes, many challenges remain, like improving bioavailability, retaining peptide stability, and endorsing efficient delivery processes for anticipated health outcomes [51,52]. This systematic review summarized evidence on the functional importance of peptides in nutrition to assess their operational features, pathways of action, and potential applications.

A key takeaway from this review is the enhanced understanding of peptide controllers for specific metabolic routes, particularly their specification in the cellular signaling process and controlling oxidative stress [65,66]. According to emerging investigations, peptides engage directly with receptors and enzymes to affect physiological responses like lipid regulations, glucose metabolism, and immunological activation [67,68,69,70]. Peptides have been demonstrated to show nutrient absorption and improvement in gut stability by enhancing functions of the epithelial cell barrier [71,72]. The latest studies in bioengineering and peptide synthesis have boosted their potential more frequently, providing precise development of synthetic peptides for therapeutic and nutritional peptides [73,74]. This research emphasizes the significance of interdisciplinary tactics incorporating molecular biology, nutritional science, and clinical research to optimize peptides in specialized healthcare and nutrition. For instance, Xue et al. (2025) developed a high internal phase emulsion by combining corn oligopeptides with chitosan, demonstrating a novel food-based functional application of peptides [75]. Our analysis is based on eight published studies that discuss peptides in nutrition that have drawn attention due to their numerous prospective advantages, including muscle recovery and adaptation, supporting extracellular matrix transformation, enhancing muscle and tendon health, and promoting overall functional performance, especially when connected with resistance training. This study shows variations in dosage methods, sample size, and evaluation criteria. The review studies demonstrate the significant function of peptides in increasing nutritional outcomes, metabolic health, musculoskeletal strength, and behavioral responses. These findings underscore the developing significance of bioactive peptides as dietary supplements for promoting health and well-being.

### 4.1. Metabolic Benefits of Peptides

The effect of peptides on Glucagon-Like Peptide-1 (GLP-1) secretion modulates the energy metabolism. Glucagon-like peptide-1 (GLP-1) fabrication is raised when protein hydrolysates are mixed with calcium-containing milk minerals, according to the most reliable investigation results. Compared with calcium citrate (CALCITR), an investigation found that GLP-1 production rose ninefold in incremental areas beneath the Curve (iAUC). Besides this effect, Peptide YY (PYY) production increased twofold, and that of Gastric Inhibitory Polypeptide (GIP) expanded 21-fold. Hormones produced within the gut, called GLP-1, GIP, and PYY, are essential in regulating hunger, glucose homeostasis, and energy metabolism. These results demonstrate that protein hydrolysates and milk minerals generate a strong hormone stimulus that might decrease appetite and raise satiety. This finding is further supported by the fact that those who ingested milk minerals showed significantly lower hunger than those who consumed CALCITR [55,76]. Zhou et al. [77] revealed that a specific food-derived peptide exhibits specific antidiabetic properties by inhibiting the dipeptidyl peptidase-4 (DPP-4) enzyme, which stimulates GLP-1 secretion. This mechanism is crucial in glycemic regulation in type 2 diabetes mellitus by increasing glucose absorption. In the same way, Zhang et al. [78] reported that multiple peptides derived from fish protein, soybean, and milk inhibit angiotensin-converting enzyme (ACE) and produce antihypertensive effects. An artificial intelligence screening methods were used to find specific peptides with ACE-inhibitory behavior.

According to particular research, there is a 21% increase in the utilization of energy when milk minerals and protein are mixed. A rise in calorie intake is a positive result for individuals who want to regulate their appetite or lose weight. This impact might be enhanced by increased GLP-1 and PYY secretion, which boosts energy metabolism while encouraging satiety. Furthermore, the observed decrease in diastolic blood pressure implies additional positive cardiovascular aspects, even though the main emphasis was on hormone response. These results coincide with other studies showing that stimulation of the GLP-1 receptor may have antihypertensive effects, potentially via accelerated vasodilation and endothelial function [55,79,80]. GLP-1 receptor activation improves endothelial function by elevating nitric oxide generation, which causes the relaxation of muscle cells in blood vessels. This relaxation dilates the vessels and enables blood to move more easily through the bloodstream. The average pressure across the veins is reduced, which drops blood pressure [81,82]. Recent clinical research explains the metabolic benefits of liraglutide when combined with physical exercise, maintaining 5.6 kg weight loss and reducing 2.3% body fat compared to liraglutide alone (GLP-1 receptor agonist), resulting in 6 kg less weight regain. This evidence highlights the crucial role of peptides in metabolism when integrating them with behavioral techniques. The dual benefit arises through a synergetic mechanism; GLP-1 receptor agonists reduce appetite while resistance training causes calories to burn during weight loss. In post-treatment, exercise preserves muscle mass by sustaining metabolic rate. When combined with resistance training, liraglutide synergistically promotes long-term biological advantages [62].

#### 4.1.1. Variations in Efficacy Among Different Treatments

Yet, there was a slight variation regardless of these encouraging findings. For instance, supplementing milk minerals with protein in one study boosted GLP-1 iAUC by up to 25% compared to protein separately [55]. However, in another study, there was no statistically significant variation between the two forms of therapy. Comparing the milk minerals to the control population in another trial, no significant increases in GLP-1, GIP, and PYY occurred, although PYY levels had elevated baseline values before protein therapy. Baseline hormone concentrations, population factors, or variations in the design of studies could all be responsible for these discrepancies. For example, a ceiling effect limited the apparent impact of milk calcium and protein supplementation in a particular investigation due to individuals having elevated baseline PYY levels. Future research should consider baseline variations when understanding mechanisms and variables regulating hormone reactivity [55,76,83,84,85].

#### 4.1.2. Implications for Nutritional Approaches

The results collectively highlight the potential of combining protein hydrolysates with calcium-containing milk minerals as a dietary strategy for managing energy balance and supporting metabolic health. These findings are particularly relevant in addressing obesity and metabolic syndrome, where appetite regulation and energy expenditure are critical targets. Hydrolysis of protein and milk minerals operates in concert to improve physiological results, which emphasizes the significance of nutritional combinations [85]. The protein hydrolysate’s anabolic and satiety-inducing features may be enhanced by calcium’s function in regulating the gut generation of hormones, producing an impact that is more apparent than either ingredient alone [86]. By adopting dietary treatments to support these processes, nutritionists can formulate an eating plan that improves daily life activities, staves off chronic health conditions, and decreases the risk of cardiovascular diseases [87,88].

### 4.2. Enhancement of Musculoskeletal Health

Collagen peptides are a potential dietary supplement to improve musculoskeletal health. Studies consistently illustrate their effectiveness in increasing fat-free mass, reducing fat mass, and building muscle strength when combined with resistance exercise [56,58]. These results highlight collagen peptides’ anabolic characteristics, which render them an effective intervention for athletes and novices. Further supporting this, Jerger et al. [60] demonstrated that a 70% increase in patellar tendon length and a 60% increase in the tendon’s cross-sectional area are enhanced by a collagen diet followed by 14 weeks of resistance training. The possible reason for this increment in length and strength of tendons is the direct involvement of collagen peptides (I, III) in the patellar tendon’s extracellular matrix (ECM) remolding and upregulating anabolic signaling supported by intense resistance training [59]. For instance, a recent investigation by Genc et al. [59] showed evidence that collagen peptides’ structural advantages lead to therapeutic effects in articular tissues. The results of this protocol show that 8-week collagen (I, III) curative potential enhances physical functions and reduces pain in meniscopathy patients. The collagen peptides (I, II, III) enforce the repair of damaged structures and inhibit cartilage-degrading enzymes, such as matrix metalloproteinases (MMPs), which cause tissue damage. This mechanism results in a 40% reduction in pain and a 25% improvement in function among patients with meniscopathy [60]. Notably, findings regarding improvements in waist circumference and quantity of bone minerals offer a distinct dimension, revealing that peptides improve skeletal health, support muscle adaptation, and contribute to body composition [89,90]. This research also revealed a synergistic effect in which peptides act as catalysts for enhancing physical strength. The analysis of collagen and whey protein brought out similar advantages, emphasizing the versatility of peptides as a protein option in dietary supplements [91].

### 4.3. Effect on Mental and Physical Health

Collagen peptides supplementation (10–20 g/day) for the long term (6–9 months) produced significant gains in daily activities, mental health, and physical health [56]. This research underscores the impact of peptides used to treat chronic pain and improve life quality, particularly for individuals with more physical activity. The ability of Peptides to reduce inflammation and support tissue repair mechanisms has a critical role in mental and physical recovery [92].

Studies show that supplementing with 10–20 g of collagen peptide (CP) per day for 6–9 months has considerable beneficial effects on pain management, mental and physical wellness, and activities of daily living (ADLs). The physical component score (PSC) in females was markedly improved by a greater dosage of 20 g daily (*p* = 0.013). In contrast, a dosage of 10 g per day was effective in enhancing ADLs (*p* = 0.031) and lowering the intensity of the pain (*p* = 0.037). The fact that the individuals receiving a placebo did not experience these beneficial effects shows the effectiveness of CP supplementation. Moreover, the 10 g per day supplement had a beneficial long-term effect on mental wellness ratings, indicating that it may help promote general well-being, particularly among those who participate in more physical activity. There are plenty of reasons for the observed disparities in effectiveness, especially in terms of physical health outcomes. The varying advantages across 10 and 20 g of CP supplements per day point to an influence by dose-response. The most significant prescription had an enormous impact on females’ physical evaluations, possibly related to variations in physiological processes such as hormonal interactions with collagen manufacturing and metabolism [57,93].

Furthermore, since people who exercised more regularly appeared to get more from the supplementation, distinctions between baseline physical activity levels directly affected the results. The fact that the placebo group revealed no discernible gains highlights the special bioactivity of collagen peptides in improving mental and physical wellness. For researchers to maximize collagen peptide supplementation concerning health advantages, future studies should look deeper at the relationship among dosage, gender-based reactions, and activity levels.

### 4.4. Influence on Taste Perception and Behavioral Responses

An interesting potential of peptides is their influence on taste perception and behavioral response. This research demonstrated a variable effect of GLP-1 infusion on taste perception in both males and females, demonstrating substantial physiological differences in eating habits and metabolic responses [58]. These results open the possibility of customized dietary treatments, in which peptides could be adjusted according to gender and other distinctive characteristics to optimize metabolic and behavioral responses. For example, women exhibit significant alterations in taste perception due to the interaction of hormones or fluctuations in GLP-1 receptors. This gender-specific distinction demonstrates the possibilities for customized dietary strategies using peptides [94,95]. This approach may increase behavioral responses by modifying nutrition plans to subjective characteristics, including age, gender, and daily routine elements.

## 5. Conclusions

This Systematic review shows the numerous health advantages of peptides in nutrition and their ability to deal with global health issues like metabolic syndrome, mental and physical wellness, behavioral response, and musculoskeletal degeneration. Collagen peptides originate from diverse dietary sources and are significant in skeletal health, fat-free mass, and enhancing muscle strength, particularly when combined with resistance training. By modulating metabolic pathways, peptides strengthen gut health, food absorption, and the prevention of chronic diseases. Moreover, significant findings demonstrate the key role of protein hydrolysates and calcium-containing milk minerals in gut hormones like GLP-1, GIP, and PYY, which are critical for appetite control, energy modulation, and glucose homeostasis. Collagen peptide dosage (10–20 g/day) significantly improves mental and physical health. Collagen peptides play a pivotal role in metabolic control and musculoskeletal rehabilitation. Collagen peptides show remarkable efficacy in enhancing patellar tendon adaptation and reducing joint pain, while liraglutide coupled with training enables a long-term weight control approach. The baseline activity levels pay significant attention to gender-specific behavior, like improving physical scores in higher-dose females. An essential ability of peptides to regulate taste perception and the impactful response of behavior demonstrates novel opportunities for targeted nutritional interventions, depending on individual characteristics like physiological differences (gender and hormonal effects).

However, the reviewed research exhibits notable limitations in methodologies, including sample size, dosing period, and evaluation criteria, which complicate comparative analyses. Additionally, most of the studies concentrated on specific demographic populations, such as women or physically active persons, which restricts the generalizability of findings to diverse populations, including people with chronic diseases or elderly individuals. The long-term safety, bioavailability, and sustainability of biopeptides as nutritional treatments remain an area of concern given the predominantly short-term nature of existing studies. Future research should prioritize long-term clinical studies to assess enduring effects, safety, and potential adverse effects of peptide supplements. Expanding study populations to include diverse demographics and standardizing research methodologies will be crucial for enabling more accurate comparisons and robust meta-analyses. Furthermore, improving the stability, delivery mechanisms, and bioavailability of peptides will be pivotal in maximizing the therapeutic potential and applicability as effective dietary supplements.

## Figures and Tables

**Figure 1 ijms-26-06043-f001:**
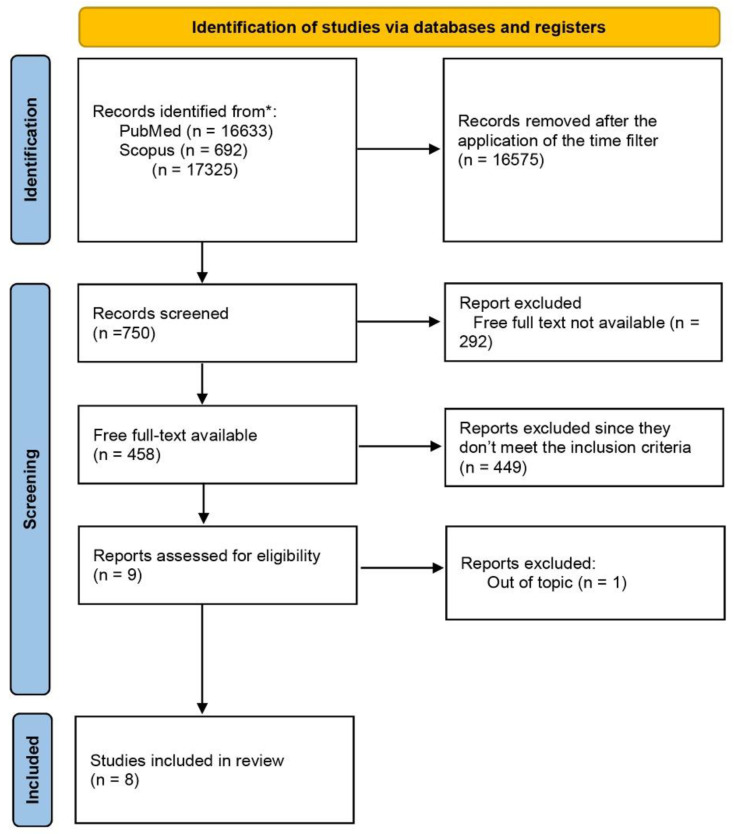
Study selection according to the Preferred Reporting Items for Systematic Reviews and Meta-Analyses (PRISMA) flow diagram. * Databases searched: PubMed (16633) and Scopus (692).

**Figure 2 ijms-26-06043-f002:**
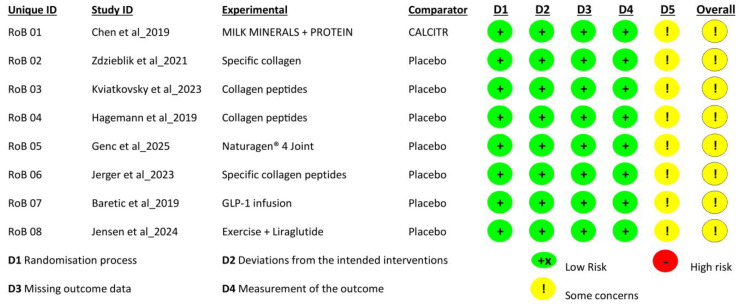
Risk of Bias Assessment using the Cochrane RoB 2.0 Tool [55,56,57,58,59,60,61,62].

**Table 1 ijms-26-06043-t001:** Search terms used for the systematic literature search.

Search Category	Phrases and Keywords Used
Peptides related terms	Peptides, nano peptides, short peptides, Biopeptides, Bioactive peptides, food-derived peptides
Nutrition-related terms	Nutrition, Supplements, Diet
Fields searched	Title, Abstract
Search operators	AND, OR
Database	PubMed, Scopus

**Table 2 ijms-26-06043-t002:** Inclusion and Exclusion Criteria for Study Selection.

Inclusion Criteria	Exclusion Criteria
Clinical Trial	Other Publication Type
Random Controlled Trial	
Article included for the first time	Duplicate
Articles published between 2019 and 2024	Articles published before 2019
Human studies	Animal, in vitro, and silico studies
Free full-text articles	Full-text to buy
Articles published in English	Articles published in other languages
	Off-topic articles

**Table 3 ijms-26-06043-t003:** Selected published clinical studies for the review.

**Author**	Study Methodology	Measuring Parameters	Findings	Summary
Chen et al. 2019 [55]	Three randomized crossover trials with healthy adults were studied to examine the effects of calcium sources and their co-ingestion with protein hydrolysate on energy consumption, appetite, and gut hormone responses. Participants were treated with calcium citrate, milk minerals with hydrolysate protein, and milk minerals alone. Blood samples were collected every 120 min.	Baseline; GLP-1, GIP, PYY, Visual analogues scales (VAS); Plasma glucose pressure, blood pressurePrimary: Incremental area under the curve (iAUC)Secondary: Indirect calorimetry, respiratory exchange ratio (RER)	Study 1: Milk Minerals plus Protein enhanced GLP-1 iAUC nine times compared with Milk Minerals and CALCITR. GIP increased 21-fold, and PYY two times in milk minerals+ protein Milk minerals suppress appetite more effectively than CALCITR Energy consumption was enhanced by up to 21% by milk minerals and protein supplementation. Study 2: GLP-1 iAUC increased by up to 25% compared to proteins. Milk minerals + protein and protein only have no notable distinction. A High PYY baseline is marked in the protein treatment group. Study 3: There is no discernible change between milk minerals and the control group in GLP-1, GIP, or PYY. GLP-1 baseline is higher in Milk Minerals treatments.	GLP-1 secretion is significantly increased by protein hydrolysate with calcium-containing milk minerals, increases energy consumption, and mildly reduces diastolic blood pressure in adults compared to milk minerals and calcium citrate.
Zdzieblik et al. 2021 [56]	In this study, three groups of overweight men received either peptide supplements, whey proteins, or placebo supplements. They followed a 12-week training program that involved daily supplement intake.	Body composition was measured by dual-energy X-ray Absorption Spectroscopy (DXA), waist circumference and body weight were measured by Anthropometric measurements, and creatine kinase and urea levels were used as blood parameters.	The study showed that, compared to the placebo group, the collagen peptide group observed an increase in fat-free mass (3.42 ± 2.54 kg) and a decrease in fat mass (−5.28 ± 3.19 kg). Muscle strength improved in all participants, with the collagen peptide group bearing the highest increment (168 ± 189 N). This study concluded that bioactive collagen peptides are more efficient when bound with resistance training than a placebo.	Fifteen grams of collagen peptides significantly decreased fat mass compared to placebo in untrained men. Bioactive collagen peptide also enhanced muscle strength in all participants. Collagen peptides and whey proteins show similar behavior. Bone mineral content and waist circumference increased in all participants.
Kviatkovsky et al. 2023 [57]	In this study, a placebo-controlled, double-blind, randomized test was conducted. All participants were divided into placebo, collagen peptide (10 g/day), and collagen peptide 20 g/day groups over 3, 6, and 9 months to measure health, pain, and physical function.	Three-day food record, physical activity survey, Veterans Rand 12 Item Health Survey (VR-12), and Knee Injury and Osteoarthritis Outcome Score (KOOS)	ADLs (*p* = 0.031) and Pain (*p* = 0.037) were enhanced by 10 g/day of collagen peptide supplements, and Physical component score (PSC) in females (*p* = 0.013) was improved by 20 g/day CP. 10 g/day of collagen peptides over 3–9 months also enhanced mental component score (*p* = 0.17). No notable improvements were observed in the placebo group.	This study indicates that collagen peptide supplementation (10–20 g/day) for 6–9 months is effective in improving daily living, physical and mental health, and pain. A daily dose of 10 g was particularly effective for individuals who exercise more frequently.
Hagemann et al. 2019 [58]	This research involved a 12-week, randomized, double-blind, placebo-controlled study. The first group received 15 g of collagen hydrolysate, and the second group received a placebo. All participants engaged in resistance training three times a week.	A bioelectrical impedance analysis system is used to measure body mass, fat-free mass, and fat mass. A Dynamometer is used to measure leg extension maximal voluntary isometric contraction (MViC), and liquid chromatography-mass spectrometry (LC-MS) is used for muscle proteome analysis.	Muscle proteome analysis showed that the collagen group contained 221 more abundant proteins, primarily related to contractile fibers, compared to only 44 in the placebo group. Significant elevations were observed in the collagen group in the pathways related to cell cycle, immune response, protein metabolism, and muscle contraction.	Collagen peptide dosage combined with resistance training significantly enhances fat-free mass, body mass, and muscle strength compared to training without these supplements.
Genç et al. 2025 [59]	This study concluded a double-blind, controlled clinical trial that involved 32 individuals divided into two groups. In this controlled study, one group was treated with collagen supplements while the other received a placebo for 8 weeks.	Physical function tests, such as Timed Up and Go (TUG), a 6 min walk, the Berg balance scale (BBS), and a stair-climbing Test, were carried out. The Tampa Scale was used for Kinesiophobia (TSK). The Visual Analog Scale (VAS), WOMAC total, KOOS-PS, and Foot Function Index (FFI) were used to measure pain, quality of life, and physical function.	SAfter eight weeks, a first group treated with collagen had significant gains in Kinesiophobia, quality of life and physical functions, and pain. In addition, leg strength is also remarkably enhanced. The placebo group exhibited no substantial changes in any studied behavior.	In meniscopathy patients, collagen enhances pain and quality of life functions. In the early stage of meniscus injury, collagen supplements might be an effective nonsurgical option for addressing symptoms.
Jerger et al. 2023 [60]	This study was a double-blind, randomized clinical trial. All participants completed 14 weeks of resistance training. One group received a daily dose of 5 g of placebo, while the other received 5 g of specific collagen peptides.	Magnetic resonance imaging (MRI) measures the Patellar Tendon and the rectus femoris muscle Cross-Sectional Area (CSA). The stiffness evaluates the patellar tendon’s stiffness. 1-repetition maximum (1 RM) tests assess maximal muscle strength.	In the collagen group, a remarkable outcome indicates a 60% (+11.4%) increase in patellar tendon CSA and 70% (+12.3%) elevation in tendon length in contrast with placebo (+4.6% and +6.1%). In both groups, 1 RM strength (20–30%) and tendon stiffness CSA (7–8%) values showed similar enhancement.	This trial summarizes that the patellar tendon cross-sections are increased significantly in the collagen group, particularly in the proximal area of the tendon, in contrast with the placebo group. However, similar outcomes were reported in tendon stiffness and muscle strength.
Baretić et al. 2019 [61]	This research involved a double-blind and placebo-controlled crossover study. All subjects were treated with Glucagon-Like-Peptide-1 (1.5 pmol/kg/min) and a placebo (0.9% saline). At the end, participants had to choose their favorite foods from the list of five tastes.	A bioelectrical impedance analyzer was used for body composition analysis. A Homeostasis Model Assessment (HOMA) calculator was used for resistance estimation and insulin sensitivity, and an ELISA ALPCO kit was used for calculating glucagon-like peptide (GLP-1).	Seven out of fourteen participants reported a change in their taste after the GLP-1 infusion compared to the placebo, which showed no change. In females, a positive correlation between insulin and GLP-1 infusion was found, which shows that individuals with higher insulin resistance have a higher response to GLP-1. These findings show that taste perception and insulin sensitivity are observed differently due to different hormonal effects.	This study concluded that taste perception affects both males and females differently. A Glucagon-Like Bioelectrical Impedance analysis system is used. Peptide infusion is more likely to change taste perception in women than men. All these outcomes from this study show some basic understanding of physiological differences in metabolic responses to eating habits among men and women.
Jensen 2024 [62]	This investigation involved a randomized, placebo-controlled trial in which all participants were treated with a low-calorie diet. In a second step, they were treated with liraglutide for one year.	Dual-Energy X-ray Absorptiometry (DXA) is used to measure body weight, and a cycle ergometer calculates VO2 max, and accelerometers assess physical activity level.	After one year of restricted treatment, the exercise group regained 3 kg in weight while the liraglutide group regained 9 kg. In contrast, the group that received combined treatment observed a 5.1 kg gain.	During active treatment, liraglutide contributes significantly to weight loss. However, its benefits diminish once treatment is discontinued. Regular physical activity leads to weight maintenance and other targeted benefits.

## Data Availability

All data supporting the findings of this study are available within the article and from the corresponding author upon reasonable request.

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
