# Peer review of "The Role of Peptides in Nutrition: Insights into Metabolic, Musculoskeletal, and Behavioral Health: A Systematic Review"

_ijms, 2025, doi:10.3390/ijms26136043_

Round 1
Reviewer 1 Report
Comments and Suggestions for Authors
The submitted review entitled “The Role of Peptides in Nutrition: Insights into Metabolic, Musculoskeletal, and Behavioral Health: A Systematic Review” is well-structured and provides a comprehensive analysis of the therapeutic benefits, molecular mechanisms, and bioactivities of peptides, with a particular focus on their relevance for metabolic, behavioral, and musculoskeletal health. The authors have done a good job synthesizing the current evidence on the potential of peptides derived from sources such as milk, fish, and plants, highlighting their bioactive properties, such as anti-inflammatory effects, improved muscle protein synthesis, and immune modulation. Some key findings, such as the role of GLP-1 in controlling taste perception and appetite stimulation, and the potential of collagen peptides to strengthen the musculoskeletal system, are relevant and well-supported by the literature.
There are two points that would enhance the clarity and completeness of the review. After addressing these, in my opinion the review can be accepted for publication.
- The representation of the study selection process in the manuscript is in line with the PRISMA guidelines, but I would suggest improving the organization of the figure (flow diagram) for better clarity and visual appeal. The current diagram could be enhanced by streamlining the layout and ensuring that the flow of the study selection process is easy to follow. Additionally, the diagram should be properly referenced and discussed in the main text of the manuscript. It would be beneficial to briefly mention the diagram and its purpose within the methods section, so readers are aware of its significance and can refer to it when needed.
- The manuscript includes a table (Table 1) titled "Published clinical trials selected for review," which presents detailed information about the studies included in the review, such as the author, study, methodology, population size, age, dose details, measurement parameters, and summary of results. This table, however, is not mentioned or discussed within the text. Although the authors describe the studies in general terms within the text, they do not provide the detailed information included in the table, such as the specifics of the study methodology, population details, and outcomes. To improve the clarity and fluency of the manuscript, I would suggest the following:
- Discuss the studies in more detail within the text, referring to the specific details presented in the table. This would help connect the information in the table with the larger narrative and provide more context for the findings.
- Simplify the table by including only the most relevant information in a more schematic form. For example, the table could highlight key aspects of each study (such as author, methodology, and key findings) concisely, while detailed aspects such as population size, age, and dose details could be summarized in the text.
This approach would ensure that readers can easily follow the details of the study in the text, while the table would remain a useful reference for quick access to key information.
Author Response
Please, attached you find our response to reviewer N. 1

Reviewer 2 Report
Comments and Suggestions for Authors
This manuscript primarily examines the effects of peptides on organisms, with analysis and discussion based on the content of five referenced papers. Thus, it can contribute to the research of peptides. However, I have to emphasize that the overall analysis relies on an insufficient literature reading, and some arguments lack proper literature citations where needed. Additionally, while various types of biological activities are mentioned, the discussion remains incomplete. Consequently, this manuscript resembles a popular science article rather than a comprehensive research review. The author should further expand their literature review, deepen their analysis, and derive more insightful conclusions. Moreover, more attention should be paid to the formatting and stylistic standards of academic writing.
1 Line 40–41: "Peptides are short chains of amino acids that come from the hydrolysis of proteins.". The definition of peptides is unclear and lacks a supporting reference.
2 Line 42: "such as fish, milk, and plants.". The three examples lack logical coherence, and a citation should be provided.
3 Line 48: "Recent". The paragraph break is unclear.
4 Line 57: "Recent". The formatting needs adjustment, as there is excessive spacing before this word. This issue should be checked throughout the entire manuscript.
5 Line 70: "Some exciting research has reported that taste perception hits differently in males and females.". A reference should be included to support this claim.
6 Line 108: The resolution of the figure is too low; it is recommended to replace it with a higher-resolution image.
7 Line 109: "Table 1 Selected published clinical studies for the review". Only five studies are discussed, which seems insufficient for a review article.
8 Lines 290–292: "tic", "sis of", "tility". Pay attention to formatting issues in these terms.
9 Line 335: "4.5 Limitations and Future Directions". It is recommended to move this section to the Conclusion part of the paper.
Author Response
Please, attached you find the response to reviewer N. 2

Reviewer 3 Report
Comments and Suggestions for Authors
Peptides play a crucial role in nutrition, influencing digestion, nutrient absorption, and overall health. Bioactive peptides often found in functional foods and nutraceuticals offer health benefits beyond basic nutrition. These bioactive peptides are involved in various physiological processes, including regulating hunger, supporting hormone function, and contributing to muscle growth and wound healing. One great example of such peptides is GLP-1. GLP-1 based drugs manage type 2 diabetes by improving glucose control, reducing appetite, and slowing down digestion.
The systematic review by Zakir et al. is an effort to understand the role of peptides in nutrition.
The authors used PubMed as the citation database for their literature search. Utilizing another database such as SciFinder or Scopus would have enhanced the scope of this systematic review. The language of the article is correct, but there is room for improvement. For example, on page 4, line 84, the sentence “The PUBMED database was carried out for the search” could be revised as “We used PubMed as the database for our literature search.” This type of sentence restructuring should be carried out throughout the entire article.
The relevant literature has been referenced to. I recommend the article be published after major changes made to the manuscript.
Below I list a few suggestions I believe could be considered to improve this article.
- Abstracts of article provide a brief overview of the research conducted. It is recommended to enhance the abstract by including a brief description of the methodology employed in the systematic review.
- Consider adding a table in the manuscript for the list of search terms used in developing search strategies. It would be useful for understanding the methodology used.
- A table should be added in the manuscript detailing the eligibility criteria upon which studies were chosen for the systematic review. It should consist of two columns – Inclusion criteria and Exclusion criteria. This would help a reader understand the underlying rationale of inclusion and exclusion at a glance.
- Figure 1 on page 5 of the manuscript is essentially a flow diagram. The legend of the figure should indicate it as a flow diagram. The following phrase can be added – Flow diagram showing the breakdown of the literature search results.
- The word “PUBMED” should be replaced with “PubMed”.
- Inclusion of a risk of bias (RoB) analysis is considered essential in a systematic review. Therefore, a RoB analysis should be included.
- Page 13, 5th column of the table from left: Glucagon-Like-Protein-1 should be Glucagon-Like-Peptide-1.
- Page 30 - Ref 77 and Ref 78 are from the same journal, Frontiers in Nutrition. In one case (Ref 77), the journal abbreviation has been used, while in the other case (Ref 78), the journal name was written in full. Also, the punctuation mark used after the volume number in Ref 77 should be changed. It should be written as in Ref 78 -“11, 1384112”. This inconsistency is observed throughout referencing and needs to be corrected. The authors should follow MDPI guidelines when referencing.
In some instances, it was hard to grasp the meaning of several sentences throughout the manuscript. The sentence structure should be simple and flawless. The authors can seek help from free online tools to correct the errors in grammar and sentence structure.
Author Response
Please, attached you find the response to Reviewer N. 3

Reviewer 4 Report
Comments and Suggestions for Authors
" The Role of Peptides in Nutrition: Insights into Metabolic, Musculoskeletal, and Behavioral Health: A Systematic Review" is prepared. Future trends should be completely enhanced. There are some places needing to be improved. Some comments are listed as follows.
Table 1 should be improved concisely.
The first appearance was the full name followed by the abbreviation, then the abbreviation should be used, for example, GLP.
The style of paragraph should be improved such as Part " 4.2 Enhancement of Musculoskeletal Health".
" 4.5 Limitations and Future Directions" should be further enhanced for their potential aspplication.
References should be checked carefully. Some are lack of important information like pages, volume....
Author Response
Please, attached you find the response to reviewer N. 4

Round 2
Reviewer 2 Report
Comments and Suggestions for Authors
Agree to publish.
Author Response
Thanks to the reviewer's comment we see that the paper has been considered publishable
Reviewer 3 Report
Comments and Suggestions for Authors
The revised version of the systematic review by Zakir et al. clearly depicts the appreciable modifications made to the manuscript following reviewers' comments, especially after the valuable inclusions of screening another database in the study, i.e., Scopus and the Risk of Bias assessment. Additionally, the quality of the English language has been improved.
I recommend that the article be published after making minor changes to the manuscript. The suggestions are listed below:
1) The name of the Scopus database hasn't been mentioned in the abstract (Page 1, Line 24). Please replace the phrase "PubMed (2019-2024) by following the...." with "the PubMed and Scopus databases in the time range of 2019 to 2024, following the...."
2) Page 8, Figure 2. In the second column (Study ID), please mention the corresponding reference numbers.
3) Page 8, Line 145. Replace "Figure 1" with "(Figure 1)".
4) Page 10, Line 192. Replace "Table 1" with "Table 3".
5) Page 9 section 3.1. The descriptions of the articles are not in sync with Table 3. For example, in Table 3, the works of Genac et al and Jarger et al are described after the works of Baretic et al. However, in section 3.1, the chronology wasn't maintained. Please rearrange it accordingly so that a reader can correlate it with Table 3. Additionally, the authors can add an additional column to the left of Table 3 mentioning the serial numbers. They can include these serial numbers in the description. For example, in Page 10, Line 192, the authors can replace the phrase "In another study listed in Table 1," with "In another study listed in Table 3 (#serial number),".
6) The reference articles are not organized. For example, in Page 32, reference number 55 is seen placed between reference numbers 52 and 53. Again in page 33, there is another reference designated with the same serial number 55. This inconsistency is throughout the manuscript and is not expected in a revised version. For example, in Table 3, column 1(Page 14), the reference number mentioned after Baretic et al is 61 and it doesn't correspond to the same article in the reference section (Page 32). The authors should correct the referencing throughout the manuscript. Additionally, the authors name hasn't been mentioned appropriately in the manuscript. Baretić et al has been mentioned as Baretic et al in Page 14. Genç et al has become Genac et al in Page 9 Line 165. Please go through the entire manuscript and make corrections.
Reviewer 4 Report
Comments and Suggestions for Authors
The manuscript The Role of Peptides in Nutrition: Insights into Metabolic, Musculoskeletal, and Behavioral Health: A Systematic Review” demonstrates that peptides display intriguing potential as nutrients for addressing global health challenges, including behavioral responses, aging, and metabolic syndrome. It needs to be revised carefully. Some comments are as follows.
Additional suggestions
- Keywords need to be revised carefully. There are too many keywords only referring to peptides.
- Table 3 should be concise rather than many sentences.
- All the tables should be three-wire meter.
4.As a review, the recent studies should be included such as corn oligopeptide as shown in https://doi.org/10.1016/j.foodhyd.2025.111268 or the others.
- Antidiabetic peptide and antihypertensive peptides should be reviewed on the discussion referred to https://doi.org/10.26599/FMH.2024.9420010 and https://doi.org/10.26599/FMH.2024.9420023.
- The reference styles should be kept consistent, for example, lack of pages or the journal name style.
